# Interactive Effects of Unhealthy Lifestyle Behaviors on Testicular Function among Healthy Adult Men: A Cross-Sectional Study in Taiwan

**DOI:** 10.3390/ijerph18094925

**Published:** 2021-05-05

**Authors:** Adi Lukas Kurniawan, Chien-Yeh Hsu, Jane C.-J. Chao, Li-Yin Lin, Rathi Paramastri, Hsiu-An Lee, Nan-Chen Hsieh, Shu-Fang Vivienne Wu

**Affiliations:** 1Research Center for Healthcare Industry Innovation, National Taipei University of Nursing and Health Sciences, Taipei 112, Taiwan; nchsieh@ntunhs.edu.tw (N.-C.H.); shufang@ntunhs.edu.tw (S.-F.V.W.); 2Department of Information Management, National Taipei University of Nursing and Health Sciences, Taipei 112, Taiwan; cyhsu@ntunhs.edu.tw; 3School of Nutrition and Health Sciences, College of Nutrition, Taipei Medical University, Taipei 110, Taiwan; rara.paramastri@gmail.com; 4Master Program in Global Health and Development, College of Public Health, Taipei Medical University, Taipei 110, Taiwan; 5Nutrition Research Center, Taipei Medical University Hospital, Taipei 110, Taiwan; 6Master Program in Applied Epidemiology, College of Public Health, Taipei Medical University, Taipei 110, Taiwan; jlin11025@gmail.com; 7School of Public Health, College of Public Health, Taipei Medical University, Taipei 110, Taiwan; 8Department of Computer Science and Information Engineering, Tamkang University, New Taipei 251, Taiwan; billy72325@gmail.com; 9College of Nursing, School of Nursing, National Taipei University of Nursing and Health Sciences, Taipei 112, Taiwan

**Keywords:** unhealthy lifestyle, smoking, alcohol consumption, sleeping habits, physical activity, diet

## Abstract

Recently, the role of lifestyle factors in testicular function has developed into a growing area of interest. Based on cross-sectional data on 3283 Taiwanese men, we investigated whether interactive effects of unhealthy lifestyle behaviors were associated with testicular function. The men were recruited from a private screening institute between 2009 and 2015. Lifestyle behaviors (smoking, alcohol drinking, physical activity (PA), sleeping habits, and diet) were obtained by a validated self-reported questionnaire. The men provided a semen sample and had blood drawn for sex hormone measurement. Men who smoked and drank had higher testosterone (T) levels (β = 0.81, *p* < 0.001) than those who neither smoked nor drank. Men who smoked and had high Western dietary pattern scores had higher T levels—by 0.38 ng/mL (*p* = 0.03). Those who drank and did not get enough sleep or had high Western dietary pattern scores had elevated T levels—by 0.60 ng/mL (*p* = 0.005) or 0.45 ng/mL (*p* = 0.02), respectively. Light PA and insomnia were associated with decreased T levels—by 0.64 ng/mL (*p* < 0.001). Those who smoked and drank or had light PA or had high Western dietary pattern scores had lower normal sperm morphologies (NSMs)—by 2.08%, 1.77%, and 2.29%, respectively. Moreover, drinkers who had high Western dietary pattern scores had higher sperm concentrations—by 4.63 M/mL (*p* = 0.04). Awareness and recognition of the long-term impact of lifestyle behaviors and better lifestyle choices may help to optimize the chance of conception amongst couples.

## 1. Introduction

Reports show that testicular function has significantly declined across the world in the past few decades. A recent meta-regression study reported that there was a significant decline in sperm concentration (SC), by 1.38 million/mL, between 1973 and 2011 among Western men. Their mean SC declined, on average, by 1.4% per year, while the total sperm counts (TSCs) declined by 1.6% per year, adding up to an overall decline of 59.3% [1]. In adult males, the testes are responsible for producing sperm and synthesizing testosterone (T), which is controlled by the central nervous system with follicle-stimulating hormone (FSH) and luteinizing hormone (LH) as the key signals [2]. The secretion of LH affects Leydig cells’ ability to promote T biosynthesis, while FSH acts on Sertoli cells to facilitate spermatogenesis. LH indirectly promotes spermatogenesis by increasing intratesticular testosterone [2]. Consecutively, T, estradiol (E2), which is produced from the aromatization of T, and inhibin B (a hormone produced by Sertoli cells) cause feedback inhibition of gonadotropin (FSH and LH) release [2]. 

It has been well-documented that smoking, drinking alcohol, physical activity, sleeping condition, and diets are associated with male sperm quality and hormones [3]. Previous studies with modest sample sizes reported that smoking is associated with lower semen volume and TSC, sperm motility, normal sperm morphology (NSM), and decreased T levels [4,5]. However, other studies with larger sample sizes show opposite results for T levels [6,7,8]. Similarly, conflicting results have also been found in relation to the association of moderate alcohol consumption and physical activity with testicular function [9,10,11,12]. Meanwhile, studies on sleeping duration and quality revealed that maintaining good sleeping behavior has beneficial effects on sperm quality and sex hormone parameters [13,14]. Diet itself has been found to be associated with testicular function. Recent studies similarly reported that adherence to a healthy prudent diet causes better testicular function [15,16]. Concurrently, the unhealthy Western diet has harmful effects on sperm and sex hormones [17]. Nonetheless, based on our knowledge, limited studies have been conducted in Asia using a larger sample size to investigate the interaction between each component of unhealthy lifestyle behaviors with testicular function. Thus, our study aimed to examine the interactive effects of unhealthy lifestyle factors on male sex hormones and sperm quality among healthy men in Taiwan.

## 2. Materials and Methods

### 2.1. Recruitments of Study Participants

This cross-sectional study used the database from the private health screening institute Mei Jau Health Management (MJHM) from 2009 to 2015. The MJHM provides annual health screenings to its members and has four clinic locations across Taiwan (Taipei, Taoyuan, Taichung, Kaohsiung). All participants signed a consent form authorized by the MJHM and the data were treated as highly confidential and were used for research purposes only. In total, 3283 healthy men (without any chronic diseases such as cancers, hepatitis, and cirrhosis) were collected from the database after we excluded those with hypertension and diabetes (*n* = 293), more than one entry (*n* = 2574), and without sex hormone or sperm quality data (*n* = 3126). The Taipei Medical University—Joint Institutional Review Board (TMU-JIRB N202010035) approved this study.

### 2.2. Anthropometry and Biochemistry Measurements

Bodyweight and height were measured by an autoanthropometers (Nakamura KN-5000A, Tokyo, Japan) during the health check-up. Body mass index (BMI) was calculated according to weight (kg) divided by the square of height (m). The percentage of body fat was analyzed by a body composition analyzer (Tanita TBF-410, IL, USA), while flexible tape measured waist and hip circumferences. Blood pressure was measured twice at 10 min intervals in a sitting position after resting for 5 min by a sphygmomanometer (Omron HEM-7201, Kyoto, Japan). Prior to the blood being drawn, all participants fasted for at least 8 hours and their fasting blood glucose (FBG) was analyzed (Toshiba C8000 autoanalyzers, Tokyo, Japan). Male sex hormones, including follicle-stimulating hormone (FSH), luteinizing hormone (LH), total testosterone (TT), and estradiol (E2), were measured by chemiluminescent immunoassay (Architect Abbott, IL, USA). Sperm quality, including sperm concentration (SC), total sperm motility (TSM), progressive motility (PRM), and normal sperm morphology (NSM), was analyzed within one hour after masturbation (at least 3 days of abstinence was required). SC was measured using a hemocytometer (Hauser Scientific Inc., Horsham, PA, USA) after being diluted in a 0.6M NaHCO_3_ solution and 0.4% (*v*/*v*) formaldehyde in distilled water [18]. WHO class A + B and WHO class A + B + C classified sperm motility as PRM and TSM, respectively [18,19]. The ten microliters of well-mixed semen were placed on a 37 °C glass slide and covered with a 22 × 22 mm coverslip. The slide was placed on the 37 °C stage microscope and examined at ×400 magnification. All the biochemistry measurements were analyzed at the MJHM central laboratory [18]. The laboratory performed rigorous quality control and calibration techniques, and thus the coefficient of variation of the samples ranged by less than 3%.

### 2.3. Lifestyle Behaviors and Other Covariates

Data related to lifestyle behaviors, including smoking status, alcohol consumption status, physical activity (PA) type, frequency of PA, sleeping type, sleeping time, and dietary habits, were collected from all men using a validated questionnaire [20,21,22]. Smoking was categorized as nonsmoker, often inhale secondhand smoke, has quit smoking, occasional smoker, and smokes daily. Drinking alcohol was categorized as none or <1 time a week, has quit drinking, 1–2 times/week, 3–4 times/week, drinks daily. Physical activity type was categorized as none, light exercise (e.g., gardening, sweeping or mopping, golf, tai chi), moderate exercise (e.g., basketball, volleyball, table tennis, badminton), heavy exercise (e.g., jogging 8 km/hours, mountain climbing, freestyle or backstroke swimming), and intensive exercise (e.g., running 12 km/h, rope-jumping, rowing, butterfly swimming). The frequency of PA was categorized as none, 1–2 h/week, 3–4 h/week, 5–6 h/week, and >7 h/week. Sleeping type was categorized as hard to fall asleep, difficulty maintaining sleep, feeling of nonrestorative sleep, use of sedatives or sleeping pills, and no problem to sleep well. Sleeping time was categorized as <4 h, 4–6 h, 6–8 h, and >8 h. Dietary servings and frequency (e.g., 5 response options per week or per day) were assessed using a standardized and validated food frequency questionnaire (FFQ) with 22 food groups, as described previously [23]. The initial questionnaire comprised 85 closed-ended questions on individual food items and was further classified into 22 nonoverlapping food groups on the basis of presumed health effects and similarity [20]. Dietary pattern was generated using principal component analysis (PCA) with orthogonal varimax rotation and based on the cut-off value of factor loading ≥ |0.30|. Two dietary patterns were identified as the prudent dietary pattern and the Western dietary pattern, with percentages of variance being 13.31% and 14.20%, respectively. The Western dietary pattern was characterized by frequent consumption of eggs, meats, inner organs, rice or flour products cooked in oil, jam or honey, sugary beverages, deep-fried foods, preserved vegetables or processed meat/fish, instant noodles, and dipping sauce (Appendix A). Dietary scores were calculated by summing up the frequency intake of food groups (1 to 5) weighed by their factor loadings, then divided into tertiles of different consumption indicating low, moderate, and high intakes. Meanwhile, other covariates included in our study were age groups (18–30, 31–40, >40 years), marital status (single, widowed, divorced, and married), education level (primary school, high school, and university), and yearly income (less than NTD 800,000 and more than NTD 800,000).

### 2.4. Statistical Analysis

Statistical analysis was performed using STATA version 13.0 (StataCorp LP, TX, USA). Categorical and continuous data were presented as number (percentage) and mean (standard deviation), respectively. The general linear model was used to determine the mean differences and a 2-sided chi-square test was performed to determine the characteristic differences among categorical data. An adjusted multivariable linear regression with 2-way interaction was used to assess the association between lifestyle behaviors and sex hormones and sperm quality. The model was analyzed to produce the beta coefficients (β) with a 95% confidence interval (CI) and adjusted by age, BMI, FBG, marital status, education level, yearly income, sleeping type, sleeping time, physical activity type, smoking status, and alcohol drinking status. Sperm quality parameters were treated as continuous variables instead of categorical variables due to only a few men having abnormal sperm quality based on the cut-off defined by the WHO criteria [19]. 

Each lifestyle behavior was dichotomized: smoker vs. nonsmoker, drinker vs. nondrinker, no/light vs. moderate/intensive PA, and low/moderate intake vs. high intake Western diet. Due to more than half of the participants not completing the question regarding the frequency of PA, we decided to exclude this variable as a part of lifestyle behavior. Moreover, we defined sleeping time as “enough” if the participants had ≥6 h of sleep duration and “not enough” if otherwise. Sleeping type was defined as “well” if the participants reported no problem sleeping well and “insomnia” if otherwise, as described previously [24]. A *p*-value < 0.05 was considered statistically significant.

## 3. Results

### 3.1. Characteristics of the Participants

Table 1 and Table 2 shows the characteristics of participants according to lifestyle behaviors. Smoker men were heavier, had a higher percentage of body fat (23.6% vs. 24.4%), WH ratio (0.84 vs. 0.85), FBG (98.7 mg/dL vs. 99.6 mg/dL), LH (3.1 IU/L vs. 3.5 IU/L), and TT (4.9 ng/mL vs. 5.3 ng/mL) but were lower in systolic BP (119.1 mmHg vs. 117.9 mmHg) and NSM (67.4% vs. 66.1%) compared with nonsmokers. Men who were drinkers had higher BMIs, percentage body fat, WH ratios, and systolic and diastolic BP, FBG, and TT (4.9 ng/mL vs. 5.3 ng/mL) levels compared to nondrinkers. In contrast, men who engaged in moderate/intense PA had lower percentage body fat, WH ratios, diastolic BP, and FBG but were higher in TT (4.9 ng/mL vs. 5.3 ng/mL) and NSM (66.4% vs. 67.6%) compared with men in the no/light PA group. There were no significant differences between sleeping type and time status with both sex hormones and sperm quality. Men who had high Western diet pa had lower NSMs (67.5% vs. 66.0%) compared to men who had low/moderate Western diet pattern scores.

### 3.2. Lifestyle Behaviors, Sex Hormones, and Sperm Quality

The association of sex hormones and sperm quality with lifestyle behaviors are shown in Table 3 and Table 4. A significant positive association with TT levels was observed for smoking status, alcohol drinking, and PA type. The adjusted model revealed that men who were smokers and drinkers were positively associated with an increase in TT levels (β = 0.36 ng/mL and 0.46 ng/mL; *p* all < 0.01, respectively). In contrast, men who had no/light PA and sleeping problems (insomnia) were negatively associated with TT levels (β = −0.41, 95% CI: −0.62, −0.19 and β = −0.23 ng/mL, 95% CI: −0.44, −0.01, respectively). Meanwhile, only smoking status had a significant association with other sex hormones (smoker on FSH: β = 1.25 ng/mL; smoker on LH: β = 0.67 ng/mL; *p* all < 0.05). There was no significant association between lifestyle behaviors with sperm quality, except Western dietary pattern with NSM. Men who had high Western dietary pattern scores were associated with reduced the percentage of NSM by 1.40% (95% CI: −2.61, −0.19; *p* = 0.023).

### 3.3. Interactive Effects of Lifestyle Behaviors on Sex Hormones and Sperm Quality

Table 5 indicates the adjusted beta coefficients of sex hormones by the interaction of lifestyle behaviors. Men who smoked and drank were more likely to have higher TT levels (β = 0.81, 95% CI: 0.51, 1.10; *p* < 0.001) than those who neither smoked nor drank. Similarly, men who smoked and had higher Western dietary pattern scores had higher TT levels, by 0.38 ng/mL (95% CI: 0.04, 0.71; *p* = 0.03), than those who neither smoked nor had a high intake of the Western diet. Men who smoked and had insomnia had reduced E2 levels (β = −2.06, 95% CI: −4.09, −0.02; *p* = 0.048), while those who smoked and did not get enough sleep time were associated with increased LH levels—by 0.89 IU/L (95% CI: 0.14, 1.64). Those who drank and did not have enough sleep or had high Western dietary pattern scores were associated with increased TT levels—by 0.60 ng/mL (95% CI: 0.18, 1.02; *p* = 0.005) or 0.45 ng/mL (95% CI: 0.06, 0.83; *p* = 0.02), respectively. Those who had no/light PA and insomnia or had high Western dietary pattern scores were associated with decreased TT levels—by 0.64 ng/mL (95% CI: −0.94, −0.34; *p* < 0.001) or 0.37 ng/mL (95% CI: −0.72, −0.02; *p* = 0.037), respectively. Additionally, those who had insomnia and had high Western dietary pattern scores had decreased E2 levels—by 2.28 IU/L (95%CI: −4.46, −0.11; *p* = 0.04).

Table 6 presents the adjusted beta coefficients of sperm quality by the interaction of lifestyle behaviors. Smokers who were drinkers or undertook no/light PA or had high Western dietary pattern scores had lower percentages of NSM—by 2.08% (95% CI: −4.02, −0.15; *p* = 0.035), 1.77% (95% CI: −3.46, −0.08; *p* = 0.04), and 2.29% (95% CI: −4.09, −0.48; *p* = 0.013), respectively—compared to nonsmokers who did not drink or undertook moderate/intense PA or had low/moderate Western dietary pattern scores. Men who undertook no/light PA and had high Western dietary pattern scores were associated with a lower percentage NSM (β = −2.08, 95% CI: −3.77, −0.39; *p* = 0.016). Likewise, those who did not sleep enough and had high Western dietary pattern scores were more likely to have a lower percentage NSM (β = −2.61, 95% CI: −4.80, −0.42; *p* = 0.02) than those who slept enough and had low/moderate Western dietary pattern scores. Additionally, only the interaction between alcohol drinking and Western dietary pattern had a significant association with SC. Drinkers who had high Western dietary pattern scores had higher SCs, by 4.63 M/mL (95% CI: 0.14, 9.12; *p* = 0.043), compared to nondrinkers and those with low/moderate Western dietary pattern scores.

## 4. Discussion

In this population-based study, smoking status, alcohol drinking, sleeping type, and type of physical activity were significantly correlated with the TT level. On the other hand, other sex hormones, including FSH and LH, were only associated with smoking status. Regarding sperm quality, NSM was found to be associated with diet. Moreover, to the best of our knowledge, our study was the first study to investigate the interactions among observed lifestyle determinants and sex hormones, as well as sperm quality.

Smoking was positively associated with increased TT concentrations in the fully adjusted model. Total T levels have been found to be higher in healthy male smokers [25,26]. A prior large-scale epidemiology study conducted using Chinese people aged 17 to 88 years suggested that smokers had significantly higher levels of TT (OR = 1.69, 95% CI: 1.34, 2.13) and free testosterone (FT) (OR = 1.27, 95% CI: 1.00, 1.61) [8]. Similarly, a meta-analysis study also reported that smokers had higher mean testosterone levels (1.53 nmol/L, 95% CI: 1.11, 1.96) than nonsmokers [27]. Furthermore, both TT and FT increased gradually as the number of cigarettes smoked increased [28]. In the current study, the levels of FSH and LH were also found to be elevated among smokers. Several studies have reported that smoking is positively associated with increased T levels by stimulating the acute release of gonadotropin-releasing hormone (GnRH) and LH; additionally, inhibition of the conversion of T to estradiol might mediate this association [7,8,29,30]. Interestingly, the elevation of T could be partially explained by the role of nicotine in cigarettes. Cotinine, a metabolite of nicotine, may act as an aromatase inhibitor, leading to increased androgens [27]. Nicotine can cross the blood–brain barrier, and thus it may stimulate the secretion of LH levels in the central nervous system [27]. Additionally, the current study found that the interactive effects of smoking with drinking, physical activity, sleeping status, and diet were negatively associated with NSM. A prior meta-analysis, using 5865 participants, also reported that declined sperm morphology was associated with frequent smoking, which was shown with the mean differences (MDs) for mild smoking (MD: −0.9%; 95% CI: −1.68, −0.12), moderate smoking (MD: −2.47%; 95% CI: −3.31, −1.64), and heavy smoking (MD: −4.24%; 95% CI: −5.02, −3.46) [31]. The chemical compounds in cigarettes, such as nicotine and cotinine, have been proposed to have detrimental effects on germ cells [29]. However, the short- and long-term effects of smoking on testicular function remain unclear.

Furthermore, our study indicated that active alcohol drinkers had higher TT levels, but insignificant results were found for sperm quality parameters. In a previous observational study among 1221 young Danish men, alcohol intake was found to be associated with increases in serum testosterone [9]. In line with the prior study, a study of 8344 healthy men from Europe and the USA showed that higher total T levels were found in the young men and fertile men groups with an alcohol intake >20 units compared to men with an alcohol intake of 1–10 units [32]. However, the authors found no consistent association between sperm quality and alcohol consumption [29]. Similarly, a study conducted in China also found no effect of alcohol use on sperm parameters [33]. In contrast, a clinical study conducted in 2005 showed the opposite results. Men who consumed a minimum of 180 mL alcohol per day ≥5 times per week showed a significant decrease in both testosterone levels and sperm quality [34]. We hypothesized that these effects may only be seen among men with long-term exposure to high levels of alcohol, while the men in our study had a relatively low frequency of alcohol consumption (64.5% of men in the alcohol group drunk alcohol 1–2 times/week). Previous studies, as discussed in a review, on alcohol consumption and sperm quality have shown inconsistent results [35]. The explanation for these differences has not been fully elucidated and further prospective investigations are needed to determine the effect of alcohol consumption on sperm quality. 

Insomnia is one of the common sleep disorders in the general population and has been recently correlated with a wide range of issues, including reproductive health. Our results demonstrated that NSM was significantly associated with the interaction between not enough sleep and high Western diet pattern score, while low T levels were found to be associated with insomnia but not with sleeping duration. An experimental study among undergraduate students aged 18–30 years reported that the sleep-deprived group had lower T levels compared with the normal sleep group—by 27% [36]. Several studies also reported that poor sleeping quality had a negative association with sperm quality, including NSM [37,38]. Similar to our study, a cohort study of 1312 men found that TT levels were unrelated to the duration of sleep [39]. Although the possible mechanism underlying the association between poor sleep and lower T levels in semen remains unclear, it is hypothesized that depression, psychological stress, and disturbance in circadian rhythm might be involved in this relationship [14]. A persistent stress condition might develop with inordinate sleep and a rise in cortisol levels. It may be hypothesized that an increment in the production of cortisol would bias the distribution of cholesterol away from T synthesis, as 17 α-hydroxy-progesterone shares part of the same route and the same intermediate substance [14]. Moreover, sleep deprivation may also cause an increase in serotonin production, which might inhibit testosterone production [40].

In the current study, no/light PA was negatively associated with TT levels. When we looked at the combined effect of PA with sleeping type and sleeping time on male sex hormones, we discovered that subjects who undertook no/light PA, regardless of their sleeping status, had significantly reduced TT levels. Meanwhile, a lower percentage of NSM and reduced TT levels were shown in the interaction between PA with a high Western diet. PA could exert both beneficial or detrimental effects depending on several inherent exercise regimen parameters, including type, intensity, and volume [41]. It has been observed that prolonged intensive exercise may lead to adverse effects on the reproductive system and fertility, such as alterations in reproductive hormone levels and atrophy of the testicular germinal epithelium, and adverse effects on spermatogenesis and changes in semen parameters, including abnormal sperm morphology and reduced sperm motility [41]. Vaamonde et al. reported improved semen parameters in physically active men when compared to sedentary people, mostly due to hormonal differences [41]. An observational study showed that moderately physically active men had significantly increased T levels compared to sedentary controls [42]. Similarly, decreased T levels were detected in men with sedentary lifestyles and activities, such as watching television [25]. Decreased oxidative stress and inflammation among men who are physically active have been proposed to explain this beneficial effect of PA [43]. 

It is well-recognized that a “healthy diet” is positively associated with sperm concentration [17]. One large clinical study, where 350 men with normal semen concentrations attended an infertility clinic, found a prudent dietary pattern (high intakes of cruciferous vegetables, leafy green vegetables, tomatoes, legumes, whole grains, fruits, fish, and chicken) to be positively associated with sperm concentration, whereas no association was observed with a “Western” dietary pattern [16]. The benefit of these “healthy” diets may be due to the high intake of antioxidants and carotenoids resulting from the foods highly consumed in the Mediterranean and similar diets. On the contrary, the “Western diet” and “sugar-sweetened drinks and snacks” diet were found to be negatively associated with SC and NSM in a study of seven-thousand young and healthy Taiwanese men [18]. Our study also showed similar results—i.e., that high consumption of Western dietary pattern was associated with reduced NSM. Furthermore, we discovered that subjects who undertook no or light PA and had high Western dietary pattern scores also had a significantly reduced T levels (*p* = 0.037) and NSM percentages (*p* = 0.016). Overall, the Western diet is known to have higher contents of saturated fats when compared to a prudent diet. Thus, nutritional intervention may be an important element in the treatment of male infertility related to abnormal sperm parameters. 

There are several limitations in our study that warrant being mentioned. First, the cross-sectional study design limited our ability to distinguish the causality of the observed relations. Second, the lifestyle factors were obtained by a self-reporting questionnaire, leaving a chance of misclassification. Due to more than fifty percent of subjects not filling in the duration and frequency of PA information, the likelihood of achieving a more robust definition of PA is limited. Moreover, there were limited data regarding the number of cigarettes consumed and duration of smoking and drinking behaviors in our study. We also did not consider alcohol consumption during the weekends, which may affect the daily consumption values. Additionally, in the validated questionnaire, the options for drinking alcohol were based on weekly units. Measuring daily alcohol units may provide a more reliable measurement than a weekly one. Third, there is no information on whether the men in the present study were aware of their fertility statuses, and thus it is likely that we introduced systematic bias. Additionally, the semen self-home collection tool may not have a similar quality as an on-site collection, and testicular volume, free testosterone, and sex hormone-binding globulin (SHBG) measurements are not available in the present study. Fourth, there is no information regarding drug use or treatment, such as anabolic androgen steroids, which may be an important contributing factor for reproductive health. Lastly, although we have adjusted our findings with some potential confounders, several confounding factors that we were unable to measure may affect our findings, including mental health status, prolonged exposure to radioactive or heavy metals, and environmental pollution. The present study had some strengths. First, unlike other studies, our study is the first study to investigate the interactive effects of lifestyle behaviors on testicular function. In addition, our study had a relatively large sample size of healthy men and we included several varieties of lifestyle risk factors.

## 5. Conclusions

In conclusion, our study suggests that modifiable behaviors, including smoking, alcohol drinking, physical activity, sleeping quality, and diet, may affect testicular function in healthy adult men. Moreover, our study also investigates the interactive effects within unhealthy determinants, which provides a better understanding of the combined consequences of these on testicular function. Future studies should clarify the underlying molecular mechanisms of these lifestyle interactions and testicular function in the general population. We foresee that such studies are needed to provide guidelines and enable physicians to recommend more appropriate clinical approaches regarding healthy lifestyles to prevent testicular dysfunction.

## Figures and Tables

**Table 1 ijerph-18-04925-t001:** Characteristics of men according to smoking, drinking, and physical activity type.

Variables	Smoking			Alcohol Drinking			Physical Activity Type		
	No	Yes	*p*	No	Yes	*p*	No/Light	Moderate/ Intensive	*p*
Age, *n* ^a^	3283		<0.01	3283		<0.01	3283		<0.01
18–30 y	754 (37.0)	376 (30.1)		958 (37.1)	172 (24.6)		546 (29.6)	584 (40.6)	
31–40 y	934 (46.0)	593 (47.5)		1192 (46.1)	335 (47.9)		878 (47.6)	649 (45.1)	
>40 y	347 (17.0)	279 (22.4)		434 (16.8)	192 (27.5)		420 (22.8)	206 (14.3)	
Marital status, *n* ^a^	3165		0.99	3165		<0.01	3165		<0.01
Single	958 (48.8)	564 (47.0)		1239 (49.7)	283 (42.0)		780 (43.9)	742 (53.5)	
Married	1006 (51.2)	637 (53.0)		1253 (50.3)	390 (58.0)		998 (56.1)	645 (46.5)	
Education, *n* ^a^	3255		<0.01	3255		<0.01	3255		<0.01
< university	543 (26.9)	573 (46.4)		828 (32.3)	288 (41.6)		710 (38.9)	406 (28.4)	
>university	1477 (73.1)	662 (53.6)		1734 (67.7)	405 (58.4)		1117 (61.1)	1022 (71.6)	
Year income, *n* ^a^	3133		0.52	3133		<0.01	3133		0.90
<NTD 800,000	1004 (51.8)	605 (50.6)		1325 (53.9)	284 (42.0)		904 (51.4)	705 (51.2)	
>NTD 800,000	934 (48.2)	590 (49.4)		1132 (46.1)	392 (58.0)		853 (48.6)	671 (48.8)	
BMI, kg/m^2 b^	23.9 3.3)	24.3 (3.5)	<0.01	24.0 (3.3)	24.4 (3.4)	0.01	24.1 (3.6)	24.0 (3.1)	0.23
Body fat, % ^b^	23.6 (5.3)	24.4 (5.6)	<0.01	23.7 (5.4)	24.4 (5.3)	<0.01	24.2 (5.6)	23.4 (5.0)	<0.01
WH ratio ^b^	0.8 (0.0)	0.8 (0.0)	<0.01	0.8 (0.0)	0.9 (0.0)	<0.01	0.8 (0.0)	0.8 (0.0)	<0.01
Systolic BP, mmHg ^b^	119.1 (12.9)	117.9 (12.9)	0.01	118.4 (12.8)	119.5 (13.3)	0.04	118.4 (13.1)	118.9 (12.6)	0.95
Diastolic BP, mmHg ^b^	72.6 (9.5)	72.4 (9.9)	0.52	72.1 (9.5)	74.3 (10.2)	<0.01	72.8 (9.9)	72.2 (9.3)	0.04
FBG, mg/dL ^b^	98.7 (11.0)	99.6 (14.6)	0.04	98.8 (12.5)	99.9 (12.5)	0.04	100.1 (14.8)	97.7 (8.5)	<0.01
FSH, IU/L ^b^	4.3 (2.9)	5.0 (7.4)	0.08	4.6 (6.2)	4.7 (3.1)	0.81	4.7 (3.8)	4.5 (7.5)	0.61
LH, IU/L ^b^	3.1 (1.4)	3.5 (2.9)	<0.01	3.3 (2.4)	3.3 (1.5)	0.94	3.2 (1.7)	3.4 (2.9)	0.45
TT, ng/mL ^b^	4.9 (1.6)	5.3 (1.8)	<0.01	4.9 (1.8)	5.3 (1.6)	<0.01	4.9 (1.7)	5.3 (1.8)	<0.01
E2, pg/mL ^b^	24.5 (9.3)	24.7 (8.5)	0.82	24.6 (9.2)	24.7 (8.1)	0.88	24.9 (9.4)	24.1 (8.1)	0.24
Prolactin, ng/mL ^b^	12.9 (7.9)	12.5 (7.6)	0.22	12.8 (7.8)	12.6 (7.8)	0.73	12.9 (7.4)	12.6 (8.3)	0.59
SC, M/mL ^b^	46.9 (25.3)	46.2 (25.6)	0.55	46.5 (25.4)	47.1 (25.4)	0.72	46.6 (25.4)	46.7 (25.5)	0.90
TSM, % ^b^	66.9 (11.8)	67.6 (11.2)	0.14	67.1 (11.6)	67.4 (11.4)	0.63	67.0 (11.3)	67.4 (11.9)	0.43
PRM, % ^b^	48.0 (14.8)	48.8 (14.7)	0.29	48.3 (14.8)	48.1 (14.7)	0.81	48.1 (14.6)	48.6 (15.0)	0.42
NSM, % ^b^	67.4 (13.4)	66.1 (13.2)	0.03	67.2 (13.2)	66.1 (13.7)	0.17	66.4 (13.2)	67.6 (13.4)	0.047

NTD, new Taiwan dollar; BMI, body mass index; WH ratio, waist to hip ratio; BP, blood pressure; FBG, fasting blood glucose; FSH, follicle-stimulating hormone; LH, luteinizing hormone; TT, total testosterone; E2, estradiol; SC, sperm concentration; TSM, total sperm motility; PRM, progressive motility; NSM, normal sperm morphology. ^a^ Chi-square test was used to determine the number (%) difference. ^b^ Linear regression analysis was used to determine the mean difference and standard deviation (SD).

**Table 2 ijerph-18-04925-t002:** Characteristics of men according to sleeping status and Western dietary pattern.

Variables	Sleeping Type			Sleeping Time			Western Dietary Pattern		
	Insomnia	Well	*p*	Not Enough	Enough	*p*	Low/Moderate	High	*p*
Age, *n* ^a^	3069		<0.01	3283		0.04	3283		<0.01
18–30 y	450 (33.0)	621 (36.4)		211 (30.4)	919 (35.5)		669 (30.6)	461 (42.1)	
31–40 y	628 (46.0)	802 (47.0)		342 (49.2)	1185 (45.8)		1005 (45.9)	522 (47.7)	
>40 y	286 (21.0)	282 (16.6)		142 (20.4)	484 (18.7)		514 (23.5)	112 (10.2)	
Marital status, *n* ^a^	2963		0.24	3165		0.02	3165		<0.01
Single	614 (46.9)	811 (49.1)		347 (52.2)	1175 (47.0)		948 (45.0)	574 (54.3)	
Married	696 (53.1)	842 (50.9)		318 (47.8)	1325 (53.0)		1160 (55.0)	483 (45.7)	
Education, *n* ^a^	3056		<0.01	3255		0.29	3255		0.64
< university	502 (37.0)	525 (30.9)		248 (36.0)	868 (33.8)		738 (34.0)	378 (34.8)	
>university	856 (63.0)	1173 (69.1)		441 (64.0)	1698 (66.2)		1432 (66.0)	707 (65.2)	
Year income, *n* ^a^	2931		0.76	3133		0.67	3133		<0.01
<NTD 800,000	689 (52.7)	847 (52.2)		332 (50.6)	1277 (51.6)		1020 (49.0)	589 (56.1)	
>NTD 800,000	618 (47.3)	777 (47.8)		324 (49.4)	1200 (48.4)		1063 (51.0)	461 (43.9)	
BMI, kg/m^2 b^	23.8 (3.3)	24.2 (3.3)	<0.01	24.6 (3.6)	23.9 (3.3)	<0.01	23.9 (3.2)	24.3 (3.7)	<0.01
Body fat, % ^b^	23.7 (5.4)	23.9 (5.3)	0.28	24.4 (5.7)	23.7 (5.3)	<0.01	23.6 (5.2)	24.3 (5.8)	<0.01
WH ratio ^b^	0.8 (0.0)	0.8 (0.0)	0.52	0.8 (0.0)	0.8 (0.0)	<0.01	0.8 (0.0)	0.8 (0.0)	0.47
Systolic BP, mmHg ^b^	118.7 (12.6)	118.7 (12.9)	0.91	119.3 (13.7)	118.4 (12.6)	<0.01	118.7 (13.0)	118.4 (12.6)	0.47
Diastolic BP, mmHg ^b^	72.7 (9.3)	72.1 (9.8)	0.16	73.1 (10.2)	72.4 (9.5)	0.20	73.0 (9.8)	71.6 (9.3)	<0.01
FBG, mg/dL ^b^	98.9 (13.4)	98.9 (12.2)	0.95	99.6 (12.9)	98.9 (12.4)	0.16	99.3 (12.6)	98.4 (12.3)	0.05
FSH, IU/L ^b^	4.6 (3.9)	4.9 (7.7)	0.56	5.0 (5.0)	4.5 (5.6)	0.26	4.6 (3.5)	4.6 (8.6)	0.88
LH, IU/L ^b^	3.3 (2.2)	3.3 (2.6)	0.95	3.5 (2.1)	3.2 (2.2)	0.20	3.3 (1.9)	3.4 (2.8)	0.49
TT, ng/mL ^b^	4.9 (1.6)	4.9 (1.7)	0.91	5.1 (1.6)	5.0 (1.8)	0.45	5.0 (1.7)	5.1 (1.8)	0.74
E2, pg/mL ^b^	23.7 (9.5)	25.0 (8.2)	0.08	25.1 (10.2)	24.4 (8.4)	0.32	24.5 (9.1)	24.7 (8.5)	0.77
Prolactin, ng/mL ^b^	13.0 (8.3)	12.5 (7.4)	0.20	12.6 (6.1)	12.8 (8.2)	0.66	12.7 (7.9)	12.8 (7.6)	0.89
SC, M/mL ^b^	46.9 (26.0)	46.5 (25.1)	0.73	45.6 (24.7)	46.9 (25.6)	0.38	47.6 (25.7)	46.1 (25.2)	0.19
TSM, % ^b^	67.3 (11.5)	67.1 (11.6)	0.77	66.5 (11.1)	67.3 (11.7)	0.23	67.1 (11.6)	67.2 (11.6)	0.96
PRM, % ^b^	48.5 (14.9)	48.2 (14.7)	0.64	48.1 (14.1)	48.3 (14.9)	0.73	48.2 (14.7)	48.4 (15.0)	0.79
NSM, % ^b^	67.0 (13.4)	67.0 (13.3)	0.80	65.9 (12.7)	67.2 (13.5)	0.06	67.5 (13.5)	66.1 (13.0)	0.01

NTD, new Taiwan dollar; BMI, body mass index; WH ratio, waist to hip ratio; BP, blood pressure; FBG, fasting blood glucose; FSH, follicle-stimulating hormone; LH, luteinizing hormone; TT, total testosterone; E2, estradiol; SC, sperm concentration; TSM, total sperm motility; PRM, progressive motility; NSM, normal sperm morphology. ^a^ Chi-square test was used to determine the number (%) difference. ^b^ Linear regression analysis was used to determine the mean difference and standard deviation (SD).

**Table 3 ijerph-18-04925-t003:** Adjusted beta (β) coefficient and 95% confidence intervals (CIs) of sex hormones according to lifestyle behaviors.

Lifestyle Factors	FSH, IU/L		LH, IU/L		TT, ng/mL		E2, pg/mL		Prolactin, ng/mL	
	β (95% CI)	*p*	β (95% CI)	*p*	β (95% CI)	*p*	β (95% CI)	*p*	β (95% CI)	*p*
Smoking (ref: nonsmoker)										
Smoker	1.25 (0.04, 2.47)	0.04	0.67 (0.19, 1.15)	0.006	0.36 (0.13, 0.59)	0.002	−0.35 (−1.87, 1.17)	0.65	−5.26 (−10.66, 0.15)	0.06
Alcohol drinking (ref: nondrinker)										
Drinker	−0.56 (−1.91, 0.79)	0.42	−0.34 (−0.87, 0.20)	0.22	0.46 (0.21, 0.71)	<0.001	1.32 (−0.37, 3.02)	0.12	5.34 (−0.87, 11.56)	0.09
Physical activity type (ref: moderate/intensive)										
No/light	−0.08 (−1.25, 1.08)	0.89	−0.19 (−0.65, 0.26)	0.41	−0.41 (−0.62, −0.19)	<0.001	0.86 (−0.59, 2.32)	0.25	2.89 (−2.59, 8.38)	0.29
Sleeping type (ref: well)										
Insomnia	−0.39 (−1.54, 0.75)	0.50	−0.12 (−0.57, 0.33)	0.59	−0.23 (−0.44, −0.01)	0.038	−1.67 (−3.11, −0.24)	0.02	−2.38 (−7.70, 2.95)	0.37
Sleeping time (ref: enough)										
Not enough	0.54 (−0.80, 1.87)	0.43	0.21 (−0.32, 0.73)	0.44	0.10 (−0.15, 0.35)	0.45	0.08 (−1.59, 1.75)	0.92	3.17 (−3.17, 9.51)	0.31
Western dietary pattern (ref: low/moderate)										
High	0.50 (−0.73, 1.72)	0.42	−0.11 (−0.59, 0.38)	0.67	−0.02 (−0.25, 0.22)	0.88	−0.51 (−2.05, 1.03)	0.51	2.87 (−2.53, 8.28)	0.29

FSH, follicle-stimulating hormone; LH, luteinizing hormone; TT, total testosterone; E2, estradiol. Adjusted by age, BMI, FBG, marital status, education level, yearly income, sleeping type, sleeping time, physical activity type, smoking status, and alcohol drinking status.

**Table 4 ijerph-18-04925-t004:** Adjusted beta (β) coefficient and 95% confidence intervals (CIs) of sperm quality according to lifestyle behaviors.

Lifestyle Factors	SC, M/mL		TSM, %		PRM, %		NSM, %	
	β (95% CI)	*p*	β (95% CI)	*p*	β (95% CI)	*p*	β (95% CI)	*p*
Smoking (ref: nonsmoker)								
Smoker	−0.65 (−3.14, 1.83)	0.61	0.76 (−0.36, 1.87)	0.18	1.12 (−0.31, 2.56)	0.13	−1.04 (−2.33, 0.25)	0.11
Alcohol drinking (ref: nondrinker)								
Drinker	1.87 (−1.22, 4.96)	0.23	0.44 (−0.94, 1.83)	0.53	−0.15 (−1.93, 1.64)	0.87	−0.74 (−2.34, 0.86)	0.36
Physical activity type (ref: moderate/intensive)								
No/light	0.64 (−1.64, 2.93)	0.58	0.02 (−1.01, 1.04)	0.97	−0.32 (−1.64, 1.00)	0.64	−0.84 (−2.02, 0.34)	0.16
Sleeping type (ref: well)								
Insomnia	0.28 (−1.99, 2.55)	0.81	0.18 (−0.84, 1.20)	0.73	0.45 (−0.87, 1.76)	0.51	−0.17 (−1.34, 1.01)	0.78
Sleeping time (ref: enough)								
Not enough	−1.22 (−4.13, 1.68)	0.41	−0.96 (−2.27, 0.35)	0.15	−0.69 (−2.37, 0.99)	0.42	−1.17 (−2.67, 0.34)	0.13
Western dietary pattern (ref: low/moderate)								
High	1.74 (−0.60, 4.08)	0.14	−0.35 (−1.41, 0.70)	0.51	−0.31 (−1.66, 1.04)	0.65	−1.40 (−2.61, −0.19)	0.023

SC, sperm concentration; TSM, total sperm motility; PRM, progressive motility; NSM, normal sperm morphology. Adjusted by age, BMI, FBG, marital status, education level, yearly income, sleeping type, sleeping time, physical activity type, smoking status, and alcohol drinking status.

**Table 5 ijerph-18-04925-t005:** Adjusted beta (β) coefficient and 95% confidence intervals (CIs) of sex hormones according to interaction of lifestyle behaviors.

Lifestyle Factors	FSH, IU/L		LH, IU/L		TT, ng/mL		E2, pg/mL		Prolactin, ng/ml	
	β (95% CI)	*p*	β (95% CI)	*p*	β (95% CI)	*p*	β (95% CI)	*p*	β (95% CI)	*p*
Smoking by drinking										
Nonsmoker										
Nondrinker	Ref		Ref		Ref		Ref		Ref	
Drinker	−0.32 (−2.49, 1.84)	0.76	−0.09 (−0.94, 0.77)	0.82	0.53 (0.16, 0.90)	0.005	1.64 (−1.08, 4.36)	0.23	−0.44 (−1.97, 1.09)	0.57
Smoker										
Nondrinker	1.35 (−0.06, 2.76)	0.06	0.78 (0.22, 1.33)	0.006	0.40 (0.13, 0.67)	0.004	−0.22 (−1.98, 1.54)	0.88	−0.60 (−1.47, 0.28)	0.18
Drinker	0.65 (−0.92, 2.22)	0.37	0.28 (−0.33, 0.90)	0.37	0.81 (0.51, 1.10)	<0.001	0.91 (−1.06, 2.89)	0.47	−0.12 (−1.34, 1.10)	0.85
Smoking by physical activity type										
Nonsmoker										
Moderate/intensive	Ref		Ref		Ref		Ref		Ref	
No/light	0.48 (−1.02, 1.99)	0.53	0.01 (−0.59, 0.61)	0.97	−0.43 (−0.70, −0.16)	0.002	0.60 (−1.30, 2.51)	0.53	−0.21 (−1.11, 0.69)	0.65
Smoker										
Moderate/intensive	2.07 (0.22, 3.91)	0.029	0.91 (0.19, 1.63)	0.013	0.31 (−0.03, 0.65)	0.71	−0.71 (−3.02, 1.60)	0.55	−0.89 (−2.07, 0.30)	0.14
No/light	1.18 (−0.46, 2.82)	0.16	0.44 (−0.21, 1.08)	0.18	−0.06 (−0.36, 0.24)	0.07	0.52 (−1.55, 2.58)	0.62	−0.31 (−1.36, 0.75)	0.57
Smoking by sleeping type										
Nonsmoker										
Well	Ref		Ref		Ref		Ref		Ref	
Insomnia	0.33 (−1.18, 1.84)	0.67	−0.01 (−0.61, 0.59)	0.96	−0.16 (−0.44, 0.11)	0.21	−1.06 (−2.96, 0.83)	0.28	0.84 (−0.07, 1.75)	0.07
Smoker										
Well	2.16 (0.43, 3.88)	0.015	0.81 (0.13, 1.49)	0.02	0.44 (0.13, 0.76)	0.006	0.41 (−1.76, 2.58)	0.59	0.01 (−1.01, 1.04)	0.98
Insomnia	0.8 (−0.80, 2.45)	0.30	0.54 (−0.10, 1.18)	0.12	0.12 (−0.19, 0.43)	0.51	−2.06 (−4.09, −0.02)	0.048	−0.19 (−1.31, 0.93)	0.74
Smoking by sleeping time										
Nonsmoker										
Enough	Ref		Ref		Ref		Ref		Ref	
Not enough	0.70 (−1.13, 2.53)	0.46	0.18 (−0.54, 0.90)	0.63	0.24 (−0.10, 0.58)	0.07	−0.35 (−2.64, 1.95)	0.79	−0.67 (−1.86, 0.53)	0.27
Smoker										
Enough	1.34 (−0.04, 2.72)	0.06	0.66 (0.11, 1.20)	0.018	0.44 (0.18, 0.70)	0.001	−0.58 (−2.31, 1.15)	0.56	−0.52 (−1.40, 0.36)	0.25
Not enough	1.69 (−0.22, 3.61)	0.07	0.89 (0.14, 1.64)	0.02	0.36 (−0.00, 0.73)	0.055	−0.03 (−2.41, 2.36)	0.89	−0.74 (−2.14, 0.65)	0.30
Smoking by Western dietary pattern										
Nonsmoker										
Low/moderate	Ref		Ref		Ref		Ref		Ref	
High	−0.16 (−1.76, 1.44)	0.86	−0.13 (−0.76, 0.50)	0.78	−0.07 (−0.37, 0.24)	0.68	0.56 (−1.45, 2.58)	0.50	0.28 (−0.67, 1.22)	0.56
Smoker										
Low/moderate	0.57 (−0.84, 1.98)	0.43	0.58 (0.03, 1.13)	0.04	0.32 (0.05, 0.60)	0.02	0.61 (−1.27, 2.48)	0.62	−0.28 (−1.29, 0.72)	0.58
High	2.37 (0.64, 4.10)	0.007	0.77 (0.08, 1.45)	0.028	0.38 (0.04, 0.71)	0.03	−1.30 (−3.44, 0.84)	0.23	−0.40 (−1.52, 0.71)	0.48
Drinking by physical activity type										
Nondrinker										
Moderate/intensive	Ref		Ref		Ref		Ref		Ref	
No/light	−0.14 (−1.49, 1.20)	0.84	−0.29 (−0.82, 0.24)	0.28	−0.47 (−0.72, −0.22)	<0.001	1.24 (−0.45, 2.93)	0.15	−0.16 (−0.97, 0.64)	0.69
Drinker										
Moderate/intensive	−0.65 (−2.80, 1.50)	0.55	−0.55 (−1.39, 0.30)	0.21	0.31 (−0.07, 0.69)	0.11	2.08 (−0.63, 4.78)	0.13	−0.60 (−2.01, 0.81)	0.40
No/light	−0.56 (−2.34, 1.22)	0.54	0.46 (−1.15, 0.24)	0.20	0.09 (−0.24, 0.42)	0.59	1.85 (−0.38, 4.08)	0.10	0.57 (−0.85, 1.99)	0.43
Drinking by sleeping type										
Nondrinker										
Well	Ref		Ref		Ref		Ref			
Insomnia	−0.52 (−1.85, 0.80)	0.45	−0.23 (−0.75, 0.29)	0.39	−0.26 (−0.51, −0.01)	0.043	−1.64 (−3.30, 0.03)	0.05	0.72 (−0.08, 1.52)	0.08
Drinker										
Well	−0.86 (−2.91, 1.19)	0.42	−0.58 (−1.39, 0.22)	0.18	0.39 (0.02, 0.76)	0.037	1.40 (−1.17, 3.98)	0.31	0.78 (−0.57, 2.13)	0.26
Insomnia	−0.87 (−2.60, 0.86)	0.37	−0.40 (−1.08, 0.28)	0.28	0.25 (−0.06, 0.57)	0.11	−0.37 (−2.54, 1.81)	0.54	0.01 (−1.39, 1.41)	0.99
Drinking by sleeping time										
Nondrinker										
Enough	Ref		Ref		Ref		Ref		Ref	
Not enough	0.64 (−0.96, 2.25)	0.42	0.05 (−0.57, 0.68)	0.89	0.07 (−0.23, 0.37)	0.44	0.70 (−1.30, 2.71)	0.511	−0.66 (−1.67, 0.34)	0.20
Drinker										
Enough	−0.46 (−2.01, 1.09)	0.58	−0.46 (−1.08, 0.15)	0.15	0.44 (0.15, 0.72)	0.003	1.86 (−0.09, 3.81)	0.08	−0.12 (−1.22, 0.99)	0.83
Not enough	−0.17 (−2.40, 2.06)	0.96	0.08 (−0.80, 0.96)	0.80	0.60 (0.18, 1.02)	0.005	0.54 (−2.26, 3.34)	0.86	0.28 (−1.80, 2.36)	0.79
Drinking by Western dietary pattern										
Nondrinker										
Low/moderate	Ref		Ref		Ref		Ref		Ref	
High	0.87 (−0.55, 2.30)	0.15	−0.07 (−0.63, 0.49)	0.61	−0.02 (−0.29, 0.25)	0.75	−1.07 (−2.85, 0.71)	0.44	0.09 (−0.74, 0.92)	0.83
Drinker										
Low/moderate	−0.07 (−1.75, 1.61)	0.83	−0.28 (−0.94, 0.38)	0.34	0.46 (0.16, 0.76)	0.003	0.58 (−1.52, 2.69)	0.39	−0.01 (−1.32, 1.29)	0.98
High	−0.56 (−2.55, 1.44)	0.84	−0.48 (−1.27, 0.30)	0.68	0.45 (0.06, 0.83)	0.02	1.55 (−0.95, 4.05)	0.42	0.29 (−1.17, 1.76)	0.69
Physical activity type by sleeping type										
Moderate/intensive										
Well	Ref		Ref		Ref		Ref		Ref	
Insomnia	−1.30 (−3.09, 0.49)	0.15	−0.21 (−0.92, 0.49)	0.55	−0.25 (−0.58, 0.07)	0.12	−0.42 (−2.67, 1.82)	0.71	1.10 (0.01, 2.19)	0.047
No/light										
Well	−0.86 (−2.50, 0.77)	0.30	−0.27 (−0.92, 0.37)	0.41	−0.42 (−0.72, −0.13)	0.005	2.01 (−0.04, 4.07)	0.055	0.55 (−0.41, 1.51)	0.26
Insomnia	−0.63 (−2.26, 1.00)	0.45	−0.33 (−0.98, 0.31)	0.31	−0.64 (−0.94, −0.34)	<0.001	−0.65 (−2.70, 1.39)	0.53	0.49 (−0.56, 1.54)	0.36
Physical activity type by sleeping time										
Moderate/intensive										
Enough	Ref		Ref		Ref		Ref		Ref	
Not enough	−0.95 (−3.13, 1.24)	0.40	−0.64 (−1.48, 0.21)	0.14	0.31 (−0.08, 0.70)	0.12	0.31 (−2.41, 3.04)	0.82	−0.57 (−2.02, 0.87)	0.43
No/light										
Enough	−0.61 (−1.91, 0.70)	0.36	−0.49 (−1.01, 0.02)	0.06	−0.34 (−0.58, −0.10)	0.005	0.98 (−0.67, 2.63)	0.25	0.03 (−0.78, 0.84)	0.94
Not enough	0.82 (−0.96, 2.60)	0.37	0.20 (−0.50, 0.90)	0.58	−0.33 (−0.67, 0.00)	0.054	0.80 (−1.45, 3.05)	0.49	−0.39 (−1.62, 0.83)	0.53
Physical activity type by Western dietary pattern										
Moderate/intensive										
Low/moderate	Ref		Ref		Ref		Ref		Ref	
High	1.50 (−0.44, 3.43)	0.13	0.28 (−0.48, 1.04)	0.47	−0.08 (−0.44, 0.28)	0.65	−0.49 (−2.92, 1.95)	0.69	0.12 (−1.00, 1.23)	0.83
No/light										
Low/moderate	0.30 (−1.06, 1.66)	0.67	−0.12 (−0.66, 0.42)	0.66	−0.45 (−0.69, −0.20)	<0.001	0.84 (−0.88, 2.55)	0.34	0.05 (−0.87, 0.98)	0.91
High	0.57 (−1.24, 2.37)	0.54	−0.08 (−0.79, 0.63)	0.82	−0.37 (−0.72, −0.02)	0.037	0.39 (−1.88, 2.66)	0.73	0.19 (−0.86, 1.24)	0.72
Sleeping type by Western dietary pattern										
Well										
Low/moderate	Ref		Ref		Ref		Ref		Ref	
High	2.08 (0.25, 3.92)	0.026	0.65 (−0.08, 1.37)	0.08	−0.13 (−0.45, 0.19)	0.87	0.61 (−1.71, 2.93)	0.60	0.25 (−0.73, 1.23)	0.62
Insomnia										
Low/moderate	0.32 (−1.08, 1.73)	0.62	0.12 (−0.43, 0.67)	0.58	−0.30 (−0.56, −0.04)	0.02	−1.47 (−3.24, 0.30)	0.15	0.57 (−0.35, 1.50)	0.23
High	−0.16 (−1.88, 1.57)	0.97	−0.32 (−1.00, 0.35)	0.66	−0.20 (−0.53, 0.13)	0.22	−2.28 (−4.46, −0.11)	0.04	0.53 (−0.56, 1.62)	0.34

FSH, follicle-stimulating hormone; LH, luteinizing hormone; TT, total testosterone; E2, estradiol. Adjusted by age, BMI, FBG, marital status, education level, yearly income, sleeping type, sleeping time, physical activity type, smoking status, and alcohol drinking status. Due to insignificant findings in all parameters, the interaction results of sleeping type by sleeping time and sleeping time by Western dietary pattern are not listed in the table (Appendix A).

**Table 6 ijerph-18-04925-t006:** Adjusted beta (β) coefficient and 95% confidence intervals (CIs) of sperm quality according to interaction of lifestyle behaviors.

Lifestyle Factors	SC, M/mL		TSM, %		PRM, %		NSM, %	
	β (95% CI)	*p*	β (95% CI)	*p*	β (95% CI)	*p*	β (95% CI)	*p*
Smoking by drinking								
Nonsmoker								
Nondrinker	Ref		Ref		Ref		Ref	
Drinker	0.50 (−4.16, 5.16)	0.83	1.00 (−1.10, 3.10)	0.35	0.68 (−2.02, 3.37)	0.62	−0.04 (−2.45, 2.38)	0.98
Smoker								
Not drinker	−1.11 (−3.87, 1.63)	0.43	0.94 (−0.29, 2.18)	0.13	1.40 (−0.19, 2.99)	0.08	−0.80 (−2.22, 0.62)	0.27
Drinker	1.80 (−1.93, 5.54)	0.34	0.96 (−0.72, 2.64)	0.26	0.62 (−1.54, 2.78)	0.57	−2.08 (−4.02, −0.15)	0.035
Smoking by physical activity type								
Nonsmoker								
Moderate/intensive	Ref		Ref		Ref		Ref	
No/light	0.40 (−2.40, 3.20)	0.78	−0.14 (−1.40, 1.12)	0.82	−0.17 (−1.79, 1.45)	0.84	−1.25 (−2.70, 0.20)	0.09
Smoker								
Moderate/intensive	−1.06 (−4.73, 2.62)	0.57	0.49 (−1.16, 2.14)	0.56	1.37 (−0.75, 3.49)	0.21	−1.73 (−3.63, 0.18)	0.07
No/light	0.06 (−3.21, 3.32)	0.97	0.81 (−0.65, 2.28)	0.28	0.76 (−1.12, 2.65)	0.43	−1.77 (−3.46, −0.08)	0.04
Smoking by Western dietary pattern								
Nonsmoker								
Low/moderate	Ref		Ref		Ref		Ref	
High	−3.48 (−6.41, −0.54)	0.02	0.02 (−1.30, 1.34)	0.98	0.26 (−1.44, 1.96)	0.76	−1.49 (−3.01, 0.03)	0.055
Smoker								
Low/moderate	−2.38 (−5.91, 1.14)	0.49	1.18 (−0.23, 2.59)	0.10	1.76 (−0.06, 3.57)	0.058	1.04 (−2.67, 0.58)	0.21
High	−3.59 (−7.40, 0.22)	0.95	0.19 (−1.37, 1.76)	0.81	0.48 (−1.54, 2.49)	0.64	−2.29 (−4.09, −0.48)	0.013
Drinking by Western dietary pattern								
Nondrinker								
Low/moderate	Ref		Ref		Ref		Ref	
High	1.24 (−1.34, 3.82)	0.35	−0.58 (−1.74, 0.58)	0.33	−1.12 (−2.61, 0.37)	0.14	−1.70 (−3.04, −0.37)	0.013
Drinker								
Low/moderate	0.64 (−3.33, 4.61)	0.75	−0.05 (−1.83, 1.74)	0.96	−1.98 (−4.27, 0.32)	0.09	−1.37 (−3.42, 0.69)	0.19
High	4.63 (0.14, 9.12)	0.043	0.59 (−1.43, 2.61)	0.57	1.31 (−1.28, 3.90)	0.32	−1.42 (−3.74, 0.90)	0.23
Physical activity type by Western dietary pattern								
Moderate/intensive								
Low/moderate	Ref		Ref		Ref		Ref	
High	2.24 (−1.20, 5.68)	0.20	−0.02 (−1.57, 1.52)	0.98	0.16 (−1.84, 2.14)	0.88	−2.32 (−4.10, −0.54)	0.011
No/light								
Low/moderate	0.96 (−1.91, 3.83)	0.51	0.25 (−1.04, 1.54)	0.70	0.01 (−1.65, 1.67)	0.99	−1.45 (−2.94, 0.03)	0.055
High	2.28 (−0.98, 5.55)	0.17	−0.38 (−1.85, 1.09)	0.61	−0.69 (−2.58, 1.19)	0.47	−2.08 (−3.77, −0.39)	0.016
Sleeping type by Western dietary pattern								
Well								
Low/moderate	Ref		Ref		Ref		Ref	
High	2.14 (−0.92, 5.20)	0.17	−0.25 (−1.62, 1.13)	0.73	−0.05 (−1.82, 1.72)	0.96	−1.67 (−3.25, −0.09)	0.039
Insomnia								
Low/moderate	0.62 (2.26, 3.50)	0.67	0.28 (−1.02, 1.58)	0.67	0.68 (−0.98, 2.35)	0.42	−0.40 (−1.89, 1.09)	0.60
High	1.82 (−1.56, 5.19)	0.29	−0.23 (−1.74, 1.29)	0.77	0.01 (−1.94, 1.97)	0.99	−1.42 (−3.17, 0.32)	0.11
Sleeping time by Western dietary pattern								
Enough								
Low/moderate	Ref		Ref		Ref		Ref	
High	1.16 (−1.44, 3.76)	0.38	−0.08 (−1.24, 1.09)	0.90	0.12 (−1.38, 1.63)	0.87	−1.32 (−2.67, 0.02)	0.054
Not enough								
Low/moderate	−2.63 (−6.47, 1.21)	0.18	−0.32 (−2.05, 1.41)	0.72	0.29 (−1.93, 2.51)	0.80	−0.90 (−2.89, 1.09)	0.37
High	1.51 (−2.72, 5.74)	0.48	−1.83 (−3.73, 0.08)	0.06	−1.81 (−4.26, 0.63)	0.15	−2.61 (−4.80, −0.42)	0.02

SC, sperm concentration; TSM, total sperm motility; PRM, progressive motility; NSM, normal sperm morphology. Adjusted by age, BMI, FBG, marital status, education level, yearly income, sleeping type, sleeping time, physical activity type, smoking status, and alcohol drinking status. Due to insignificant results in all parameters, the interactions of smoking and sleeping type and time and Western dietary pattern, drinking and physical activity type, sleeping type and time, physical activity type and sleeping type and time, and sleeping type and sleeping time are not listed in the table (Appendix A).

## Data Availability

The data that support the findings of this study are available from Mei Jau (MJ) Health Management Institute but are restricted to research use only. The data are not publicly available. Data are available from the authors upon reasonable request and with permission of the MJ Health Management Institute.

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
