# Peer review of "Interactive Effects of Unhealthy Lifestyle Behaviors on Testicular Function among Healthy Adult Men: A Cross-Sectional Study in Taiwan"

_ijerph, 2021, doi:10.3390/ijerph18094925_

Round 1

Reviewer 1 Report

The effects of lifestyle behaviors (smoking, alcohol drinking, physical activity (PA), sleeping habits, and diet) on testicular function are being analyzed. Revision is required for the following points.

First, the existing research results related to this study have not been sufficiently discussed. It is recommended to systematically review previous studies to see how smoking, alcohol drinking, physical activity (PA), sleeping habits, and diet each affect outcome variables. Section about theory and hypothesis is required.

Second, it is being analyzed how each causal variable affects the health outcome. It is necessary to put various independent variables into one regression analysis model and analyze how the results vary.

Reviewer 2 Report

Thank you for inviting me to review the Adi Lukas Kurniawan and colleagues paper entitled “Interactive effects of unhealthy lifestyle behaviours on testicular 2 function among healthy adult men: A cross-sectional study in Taiwan”. I appreciate the authors’ work, but I have a few concerns regarding the manuscript, which are listed below.
Lines 110-112: Can I ask the authors to indicate the details of the validation of the questionnaires used to obtain lifestyle data? Do the questionnaires have names/acronyms, authors? Was the validation document published earlier? If yes, please add the necessary references. If not, please explain/ present some validation information.
Lines 124-125: First, please specify the consumption of how many single food items FFQ covered. Information about 22 food groups, to which the consumption was reduced, is important for further analysis (PCA). Second, the reference you cited is not connected with the source of the questionnaire or its validation paper. You should turn to the right sources.
Line 128: Did the factor loadings were analysed as absolute or relative value? If absolute, please use appropriate marking/designation.
Lines 128-131: Please, add the values of the total variance explanation and a share for each of DPs. Add it also to the Table S1.
Table S1: PCA was based on 22 food groups, but in Table S1 are mentioned 18 of them. I understand that only them are included in DPs, but I and probably the readers want to know, which were not.
Table 2: Understand that sleeping time and type may be divide into insomnia and well subgroups, but why Western DP is divided in the same way? Shouldn’t it be low/moderate and high?
Line 317: This sentence doesn't fit in this place or should be rephrased.
Editing comment:
General. Tables should have the same body style, Especially this presented in the main text.
Lines 282, 292: Please, use the right one (square) brackets for citation.
Line 302: Start sentence from capital letter.
Lines 313, 315/316: Chose one form for the same variable when you mention it in the text (“no/light PA tyle” or “no or light PA”). It’ll be easier for readers.
References style should be unified (in regards to the journal names, they should be abbreviated)

Author Response

Dear reviewers,

We thank you for your comments and suggestions for improving the manuscript. The authors have considered the comments and revised the manuscript accordingly. Thank you very much.

Yours sincerely,

Adi Lukas Kurniawan, Ph.D. (corresponding author)

Reviewer 2

  1. Lines 110-112: Can I ask the authors to indicate the details of the validation of the questionnaires used to obtain lifestyle data? Do the questionnaires have names/acronyms, authors? Was the validation document published earlier? If yes, please add the necessary references. If not, please explain/ present some validation information.

REPLY: According to the MJ health management (MJHM) center, the questionnaire used to obtain lifestyle data has been validated and has been published in our previous studies [1-4]. Moreover, the details regarding the other previous publications that have been used the same questionnaire can be seen on the official website of the MJHM (http://www.mjhrf.org/main/page/resource/en/%23resource08). Additionally, we have added necessary references accordingly in our manuscript.

  1. Lines 124-125: First, please specify the consumption of how many single food items FFQ covered. Information about 22 food groups, to which the consumption was reduced, is important for further analysis (PCA). Second, the reference you cited is not connected with the source of the questionnaire or its validation paper. You should turn to the right sources.

REPLY: We have added some information regarding the FFQ as well as the references as follows:

[Line 129 – 134] Dietary servings and frequency (e.g. 5 response options per week or per day) were assessed using a standardized and validated food frequency questionnaire (FFQ) with 22 food groups as described previously [2]. The initial questionnaire comprised 85 closed-ended questions on individual food items and was further classified into 22 non-overlapping food groups on the basis of presumed health effects and similarity [5].

  1. Line 128: Did the factor loadings were analysed as absolute or relative value? If absolute, please use appropriate marking/designation.

REPLY: The factor loadings were analyzed as an absolute value, therefore we have revised it accordingly.

[Line 134 – 135] Dietary pattern was generated using principal component analysis (PCA) with orthogonal varimax rotation and based on the cut-off value of factor loading ≥ |0.30|

  1. Lines 128-131: Please, add the values of the total variance explanation and a share for each of DPs. Add it also to the Table S1.

REPLY: We have added the values of the total variance in the manuscript and Table S1.

[Line 135 – 137] Two dietary patterns were identified as the Prudent-dietary pattern and the West-ern-dietary pattern with the percentage of variance 13.31% and 14.20% respectively.

  1. Table S1: PCA was based on 22 food groups, but in Table S1 are mentioned 18 of them. I understand that only them are included in DPs, but I and probably the readers want to know, which were not.

REPLY: We have revised it and added the food groups which were not included in DPs accordingly in table S1.

  1. Table 2: Understand that sleeping time and type may be divide into insomnia and well subgroups, but why Western DP is divided in the same way? Shouldn’t it be low/moderate and high?

REPLY: Thank you to a reviewer for the correction. We have revised it accordingly

  1. Line 317: This sentence doesn't fit in this place or should be rephrased.

REPLY: We have rephrased it as follows:

[Line 333-334] Meanwhile, a lower percentage of NSM and reduced TT level were shown in the interaction between PA with a high Western diet.

Editing comment:

General. Tables should have the same body style, Especially this presented in the main text.

REPLY: Thank you to the reviewer for the comment. Based on our concern, we already follow the Table style from the manuscript template provided on the website.

Lines 282, 292: Please, use the right one (square) brackets for citation.

REPLY: We have revised it accordingly

Line 302: Start sentence from capital letter.

REPLY: We have revised it accordingly

Lines 313, 315/316: Chose one form for the same variable when you mention it in the text (“no/light PA tyle” or “no or light PA”). It’ll be easier for readers.

REPLY: We have revised it accordingly

References style should be unified (in regards to the journal names, they should be abbreviated)

REPLY: We have checked and used the references style according to the journal template

References

  1. Kurniawan, A.L.; Hsu, C.Y.; Chao, J.C.J.; Paramastri, R.; Lee, H.A.; Lai, P.C.; Hsieh, N.C.; Wu, S.F.V. Association of Testosterone-Related Dietary Pattern with Testicular Function among Adult Men: A Cross-Sectional Health Screening Study in Taiwan. Nutrients 2021, 13, doi:10.3390/nu13010259.
  2. Paramastri, R.; Hsu, C.Y.; Lee, H.A.; Lin, L.Y.; Kurniawan, A.L.; Chao, J.C. Association between Dietary Pattern, Lifestyle, Anthropometric Status, and Anemia-Related Biomarkers among Adults: A Population-Based Study from 2001 to 2015. Int J Environ Res Public Health 2021, 18, doi:10.3390/ijerph18073438.
  3. Kurniawan, A.L.; Hsu, C.Y.; Rau, H.H.; Lin, L.Y.; Chao, J.C. Inflammatory Dietary Pattern Predicts Dyslipidemia and Anemia in Middle-Aged and Older Taiwanese Adults with Declined Kidney Function: A Cross-Sectional Population Study from 2008 to 2010. Nutrients 2019, 11, doi:10.3390/nu11092052.
  4. Syauqy, A.; Hsu, C.Y.; Rau, H.H.; Kurniawan, A.L.; Chao, J.C. Association of Sleep Duration and Insomnia Symptoms with Components of Metabolic Syndrome and Inflammation in Middle-Aged and Older Adults with Metabolic Syndrome in Taiwan. Nutrients 2019, 11, doi:10.3390/nu11081848.
  5. Lyu, L.C.; Lin, C.F.; Chang, F.H.; Chen, H.F.; Lo, C.C.; Ho, H.F. Meal distribution, relative validity and reproducibility of a meal-based food frequency questionnaire in Taiwan. Asia Pac J Clin Nutr 2007, 16, 766-776.

Reviewer 3 Report

I found the paper interesting and generally well written. I have a few comments though:

  • Line 101: : it is unclear if authors quantified free or total testosterone.
  • In section 2.3 of the methodology (and in the abstract) it says that it uses a validated questionnaire, but it does not reference it nor is it clear to me. I think that the measurement method is the main problem of this study, so it should explain better the methods used to measure and why.

The rest of the article seems well written.

Author Response

Dear reviewers,

We thank you for your comments and suggestions for improving the manuscript. The authors have considered the comments and revised the manuscript accordingly. Thank you very much.

Yours sincerely,

Adi Lukas Kurniawan, Ph.D. (corresponding author)

Reviewer 3

  1. I found the paper interesting and generally well written. I have a few comments though: Line 101: it is unclear if authors quantified free or total testosterone.

REPLY: Thank you to the reviewer for the valuable feedback. The current study quantified total testosterone (TT) which have been reported in a previous study [1]. Moreover, we have revised accordingly for the entire manuscript

  1. In section 2.3 of the methodology (and in the abstract) it says that it uses a validated questionnaire, but it does not reference it nor is it clear to me. I think that the measurement method is the main problem of this study, so it should explain better the methods used to measure and why. The rest of the article seems well written.

REPLY: We have added the references regarding the validated questionnaire [2,3] and we have added more description of the measurement methods as follows:

[Line 105 – 107] SC was measured using a hemocytometer (Hauser Scientific Inc., Horsham, PA, USA) after being diluted in a 0.6M NaHCO3 solution and 0.4% (v/v) formaldehyde in distilled water [4].

[Line 108 – 111] The 10 microliter of well-mixed semen were placed on a 370C glass slide and covered with a 22 x 22 mm coverslip. The slide was placed on the 370C stage microscope and examined at x400 magnification [4]. All the biochemistry measurements were analyzed at the MJHM central laboratory.

References:

  1. Kurniawan, A.L.; Hsu, C.Y.; Chao, J.C.J.; Paramastri, R.; Lee, H.A.; Lai, P.C.; Hsieh, N.C.; Wu, S.F.V. Association of Testosterone-Related Dietary Pattern with Testicular Function among Adult Men: A Cross-Sectional Health Screening Study in Taiwan. Nutrients 2021, 13, doi: 10.3390/nu13010259.
  2. MJ Group. MJ Health Screening Center Questionnaire QR-121-1 MJ2011.06-1104TW. Available online: https://www.mjlife.com/index.aspx?lang=chi&fn=index (accessed on 3 August 2020).
  3. Lyu, L.C.; Lin, C.F.; Chang, F.H.; Chen, H.F.; Lo, C.C.; Ho, H.F. Meal distribution, relative validity and reproducibility of a meal-based food frequency questionnaire in Taiwan. Asia Pac J Clin Nutr 2007, 16, 766-776.
  4. Liu, C.Y.; Chou, Y.C.; Chao, J.C.J.; Hsu, C.Y.; Cha, T.L.; Tsao, C.W. The Association between Dietary Patterns and Semen Quality in a General Asian Population of 7282 Males. Plos One 2015, 10, doi:10.1371/journal.pone.0134224.

Round 2

Reviewer 1 Report

Authors tried to accept the comment from reviewers.